# Prevalence of Sleep Disturbance and Its Risk Factors in Patients Who Undergo Surgical Treatment for Degenerative Spinal Disease: A Nationwide Study of 106,837 Patients

**DOI:** 10.3390/jcm11195932

**Published:** 2022-10-08

**Authors:** Jihye Kim, Min Seong Kang, Tae-Hwan Kim

**Affiliations:** 1Division of Infection, Department of Pediatrics, Kangdong Sacred Heart Hospital, Hallym University College of Medicine, 150 Seongan-ro, Seoul 05355, Korea; 2Spine Center, Department of Orthopedics, Hallym University Sacred Heart Hospital, Hallym University College of Medicine, 22 Gwanpyeong-ro, Anyang 14068, Korea

**Keywords:** sleep disturbance, sleep disorder, sleep medication, epidemiology, prevalence, surgery, spine, degenerative spinal disease, risk factors

## Abstract

Spinal surgeons have not yet considered sleep disturbance an area of concern; thus, a comprehensive study investigating the epidemiology of sleep disturbance in patients with degenerative spinal disease is yet to be conducted. This study aimed to fill this research gap by investigating the epidemiology of sleep disturbance in patients who underwent spinal surgery for degenerative spinal disease and identifying the associated risk factors. This nationwide, population-based, cohort study, used data from January 2016 and December 2018 from the Korea Health Insurance Review and Assessment Service database. This study included 106,837 patients older than 19 years who underwent surgery for degenerative spinal disease. Sleep disorder was initially defined as a diagnosis of a sleep disorder made within one year before the index surgery and identified using the International Classification of Diseases, 10th revision, codes F51 and G47 (main analysis). We also investigated the use of sleep medication within 90 days prior to the index surgery, which was the target outcome of the sensitivity analysis. The prevalence of sleep disturbance was precisely investigated according to various factors, including demographics, comorbidities, and spinal region. Logistic regression analysis was performed to identify the independent factors associated with sleep disturbance. The results of the statistical analysis were validated using sensitivity analysis and bootstrap sampling. The prevalence of sleep disorder was 5.5% (*n* = 5847) in our cohort. During the 90 days before spinal surgery, sleep medication was used for over four weeks in 5.5% (*n* = 5864) and over eight weeks in 3.8% (*n* = 4009) of the cohort. Although the prevalence of sleep disturbance differed according to the spinal region, the spinal region was not a significant risk factor for sleep disorder in multivariable analysis. We also identified four groups of independent risk factors: (1) Age, (2) other demographic factors and general comorbidities, (3) neuropsychiatric disorders, and (4) osteoarthritis of the extremities. Our results, including the prevalence rates of sleep disturbance in the entire patient population and the identified risk factors, provide clinicians with a reasonable reference for evaluating sleep disturbance in patients with degenerative spinal disease and future research.

## 1. Introduction

Sleep plays an essential role in both cognitive and physiologic function [1,2]. Therefore, sleep disturbance can not only have detrimental effects on quality of life, but also potentially cause mental and physical illness, eventually increasing the risk of mortality [3,4]. Sleep disturbance is prevalent globally, and nationwide studies have revealed that more than 20% of the general population suffers from sleep disturbance [5,6,7].

Chronic pain is one of the major risk factors associated with sleep disturbance [8,9], and sleep disturbance has been reported to be prevalent in patients with degenerative joint diseases of the extremities [10,11]. Recent studies have revealed that sleep disturbance is also prevalent in patients with degenerative spinal disease, with a reported prevalence ranging from 11 to 74% [12,13,14,15]. Interestingly, studies have identified that in patients with degenerative spinal disease, the radiologic severity of degeneration is a stronger predictor of sleep disturbance than overall pain intensity [12,13]. In addition, the radiologic indices associated with sleep disturbance differed according to the spinal regions. For example, in patients with lumbar stenosis, sleep disturbance was more associated with foraminal-type stenosis than central-type stenosis [13]. In contrast, in patients with cervical myelopathy, central-type stenosis was more closely associated with sleep disturbance than foraminal-type stenosis [12]. From these results, the authors deduced that the mechanisms of sleep disturbance may differ according to the spinal regions and that sleep disturbance in patients with cervical myelopathy might be caused by the same factors causing sleep disturbance in patients with spinal cord injury, such as symptoms associated with cord injury, including pain, sleep breathing disorder, and sleep movement disorder, as well as inhibition of the neural pathway for endogenous melatonin production passing through the cervical spinal cord.

Considering that the radiologic degree of spinal degeneration is closely associated with sleep disturbance, sleep disturbance is expected to be particularly prevalent in patients who are considering surgical treatment for degenerative spinal disease, which could have influenced their choice to undergo surgical treatment. However, sleep disturbance has hitherto not been a matter of concern to spinal surgeons, and few studies have investigated the epidemiology of sleep disturbance in patients who underwent spinal surgery. Although several recent studies have been conducted for this purpose, they had the following limitations [12,13,14,16,17]. First, most of these studies are single-center studies with a limited number of patients. Thus, the prevalence rates of sleep disturbance and the estimates for their risk factors can be biased, reflecting the skewness of their study samples. Second, because of the small sample size, a comprehensive epidemiologic analysis including all spinal regions and considering various morbidities that are prevalent in patients with degenerative spinal disease could not be performed. This information would be very useful not only for clinicians, but also for researchers to understand the etiology or mechanisms of sleep disturbance in patients with degenerative spinal disease.

Our study has two distinct research purposes. First, by using a nationwide database that included the entire population, we aimed to investigate the epidemiology of sleep disturbance in patients who underwent spinal surgery for degenerative spinal disease. Based on the large dataset, the epidemiology of sleep disturbance was precisely investigated according to various clinical profiles, including demographics, various comorbidities, and spinal regions. We particularly focused on investigating the prevalence of sleep disturbance according to spinal regions, which has not been reported in previous studies due to the limited number of cases. Second, using this information, we attempted to identify independent risk factors for their sleep disturbance.

## 2. Patients and Methods

### 2.1. Database

In this nationwide population-based cohort study, data were obtained from the Korea Health Insurance Review and Assessment Service (HIRA) database. The HIRA database contains all inpatient and outpatient data from hospitals and community clinics in Korea, allowing for a nationwide cohort study that includes the entire population. Diagnostic codes were assigned according to the modified version of the 10th revision of the International Classification of Diseases (ICD-10) and the seventh revision of the Korean Classification of Diseases. Drug use under diagnosis was identified using anatomical therapeutic chemical (ATC) codes and the HIRA general name codes. This study was approved by the Institutional Review Board of our hospital (IRB No. 2020-03-009-001).

### 2.2. Study Patients

We included patients aged >19 years who underwent surgical treatment (index surgery) for degenerative spinal disease between 1 January 2016 and 31 December 2018 (Figure 1). Degenerative spinal diseases were identified using the following codes: Spinal stenosis (M48.0), spondylolisthesis (M43.1), spondylolysis (M43.0), other spondylosis (M47.1 and M47.2), and cervical disc disorder (M50).

The spinal region of surgical treatment was identified using the following electronic data interchange codes: Cervical surgery including cervical decompressive (N2491, N2492, N0491, N1491, N1497, N2497) and fusion (N2461, N0464, N2463, N2467, N2468, N0467, N2469) surgery; thoracic surgery including thoracic decompressive (N1492, N1498, N2498) and fusion surgery (N0465, N2464, N2465, N2466, N0468), and lumbar surgery including lumbar decompressive (N0492, N1493, N1499, N2499) and fusion (N0466, N1466, N0469, N1460, N1469, N2470) surgery. We excluded patients who were treated under the ICD-10 codes of spinal infection (A18.00, M46, M49, G06, and T814), spine fractures (S1, S2, S3, T02.0, T02.1, T02.7, T08, T09, T91, M48.3, M48.4, and M48.5), or malignancy (C) within two years before the index surgery (Figure 1).

### 2.3. Definitions of Sleep Disturbance

Sleep disturbance in the cohort was identified using the following two methods (Figure 2). First, sleep disturbance was primarily defined as a diagnosis of sleep disorder within one year before the index surgery. Preoperative sleep disorder was identified using the following diagnostic codes: Nonorganic sleep disorders (F51), and sleep disorders (G47). This was then used as the target outcome in the main analysis. Second, in the sensitivity analysis performed to internally validate our results, sleep disturbance was additionally defined by the use of sleep medication during the 90 days before the index surgery. Sleep medication was defined as drugs currently available for insomnia approved by the Korean Food and Drug Administration, including flurazepam, triazolam, flunitrazepam, brotizolam, zolpidem, eszopiclone, doxepin, doxylamine, and diphenhydramine [18]. Among them, antihistamines, including doxylamine and diphenhydramine, were excluded. The ATC and HIRA general name codes for sleep medication are presented in Appendix A. Data regarding preoperative sleep medication were used as the target outcome in the sensitivity analysis.

### 2.4. Factors Associated with Sleep Disturbance

Demographic data at the time of surgery were retrieved. Medical conditions diagnosed in the year before the index surgery were identified using ICD-10 codes (Appendix A) and evaluated using the Charlson comorbidity index (CCI) [19,20,21]. We also investigated neuropsychiatric disorders that were possibly associated with sleep disturbance using ICD-10 codes (Appendix A). The diagnosis of depression was confirmed using the ATC codes for the use of antidepressants (N06A, Appendix A).

We also evaluated osteoarthritis of the extremities using a validated method in our database [22]. Patients with osteoarthritis of the extremities were identified using the ICD-10 codes for osteoarthritis (M15 to M19) with corresponding radiographs of the extremities. The HIRA electronic data interchange codes for X-rays of the extremities are presented in Appendix A.

### 2.5. Statistical Analysis

Data are reported as the mean ± standard deviation for numerical variables, and as numbers and frequencies (%) for categorical variables. The prevalence of sleep disturbance was precisely presented according to the factors associated with sleep disturbance and the spinal regions. For the main analysis, sleep disturbance, defined as the diagnosis of a sleep disorder within one year before the index surgery, was chosen as the dependent variable. Logistic regression analysis was performed to identify independent factors associated with sleep disturbance, with adjustment for variables identified to be significant in the univariable analysis (*p* < 0.05).

Our statistical model was validated using the following procedures. First, a sensitivity analysis was performed to validate risk factors. Sleep disturbance was defined according to the use of sleep medication during the 90 days before the index surgery and was used as the dependent variable for the sensitivity analysis. Second, all estimates from the main and sensitivity analyses were validated using the bootstrap method. All estimates were internally validated with relative bias based on 1000 bootstrapped samples. Relative bias was estimated as the difference between the mean bootstrapped regression coefficient estimates and the mean parameter estimates of multivariable model divided by the mean parameter estimates of the multivariable model.

Multicollinearity between covariates was tested using a variance inflation factor. Data extraction and statistical analysis were performed using the SAS Enterprise Guide 6.1 (SAS Institute, Cary, NC, USA).

## 3. Results

Between 2016 and 2018, 198,844 patients underwent spinal surgery (index surgery) for degenerative spinal disease (Figure 1). Among them, we excluded patients who were treated under the ICD-10 codes of malignancy (*n* = 11,504), spinal infection (*n* = 1937), and spinal fracture (*n* = 81,463) within two years before the index surgery, and those who had missing data (*n* = 376).

A total of 106,837 patients were included in this study, with a mean age of 62.9 years and 52% (*n* = 55,595) being women.

### 3.1. Annual Prevalence of Sleep Disturbance According to the Three Definitions

Among the 106,837 patients, sleep disorders were diagnosed within one year before the index surgery in 5.5% (*n* = 5847, Table 1). During the 90 days before spinal surgery, sleep medication was used for over four weeks in 5.5% of the cohort (*n* = 5864) and over eight weeks in 3.8% (*n* = 4009) of the cohort. During the study period, the number of patients with preoperative sleep disorders and those who used sleep medications continuously increased (Table 1).

### 3.2. Prevalence of Sleep Disturbance According to the Baseline Characteristics and Comorbidities

Sleep disorders were common in patients of older age, female sex, urban residence, and surgery at a tertiary hospital (Table 2). The difference was most pronounced by age, and patients aged over 80 years had approximately three-fold higher chances of having sleep disturbance than those between 20 and 49 years (8.8% vs. 2.7%).

Patients with a sleep disorder had a slightly higher CCI score than those without it (1.56 ± 1.44 vs. 1.12 ± 1.26). However, the prevalence of sleep disorders did not show an increasing trend according to categorized CCI scores (Table 3). Conversely, patients with CCI scores ≥ 6 points had approximately one-half lower chances of having sleep disturbance than those with CCI scores ≤ 2 points (2.9% vs. 6.0%). Patients with specific comorbidity had a higher prevalence of sleep disorder than the overall prevalence (5.5%, Table 3). Sleep disorder was especially frequent in patients with neuropsychiatric comorbidities, including depressive disorder (11.8%), dementia (12.0%), Parkinson’s disease (11.4%), migraine (11.9%), tension-type headache (11.4%), and other-type headache (10.9%). Diagnosis of sleep disorder was also frequent in patients with concurrent osteoarthritis of the extremities, especially in the ankle (9.1%), wrist (8.1%), and shoulder (7.9%).

The proportions of patients who had over 4- or 8-week sleep medication during the 90 days before the index surgery were generally concordant with the proportions of those who were diagnosed with sleep disorders (Table 2 and Table 3).

### 3.3. Prevalence of Sleep Disturbance According to Spinal Regions

The prevalence of sleep disorders was 6.9%, 5.7%, and 4.4% in patients with thoracic, lumbar, and cervical spinal lesions, respectively (Figure 3). Prevalence rates of sleep disturbance defined by the use of sleep medication were also concordant with the proportions of those who were diagnosed with a sleep disorder, and the patients who underwent thoracic spine surgery consistently showed the highest prevalence rates according to all three definitions of sleep disturbance (Figure 3).

### 3.4. Prevalence of Sleep Disturbance According to Concurrent Neuropsychiatric Disorders and Osteoarthritis of Extremities

The two most common types of concurrent neuropsychiatric disorders in our cohort were depressive disorder (21.8%, *n* = 23,921) and cerebrovascular disease (8.9%, *n* = 9502; Table 3), which were more common in patients with thoracic or lumbar lesions (Table 4). The prevalence of the three types of sleep disturbance according to the spinal region and concurrent neuropsychiatric disorders are presented in Table 4. The prevalence of sleep disorder in patients with a specific neuropsychiatric disorder was higher in those with a lumbar lesion than in those with a cervical lesion.

The three most common regions of concurrent osteoarthritis in our cohort were the knee (22.8%, *n* = 24,338), shoulder (8.0%, *n* = 8503), and hip (6.6%, *n* = 7104; Table 3). Osteoarthritis of the upper extremities was the most common in patients with a cervical lesion, and that of the lower extremities was common in patients with thoracic or lumbar lesions (Table 5). We present the prevalence of three types of sleep disturbance according to spinal region and concurrent osteoarthritis of the extremities in Table 4. The prevalence of sleep disorder in patients with concurrent osteoarthritis of the upper extremities was higher in those with lumbar lesions than in those with cervical lesions.

### 3.5. Risk Factors for Sleep Disorder: Main Analysis

Multivariable analysis identified the following variables as significant risk factors for sleep disturbance in patients who underwent surgical treatment for degenerative spinal diseases: (Table 6): Age of 50–69 years (odds ratio, OR [95% confidence interval] = 1.40 [1.25–1.57]), age of 70–79 years (OR = 1.80 [1.60–2.03]), age over 80 years (OR = 2.22 [1.92–2.58]), female sex (OR = 1.14 [1.07–1.21]), urban residence (OR = 1.18 [1.09–1.27]), surgery at a tertiary hospital (OR = 1.08 [1.00–1.16]), peripheral vascular disease (OR = 1.22 [1.13–1.32]), chronic pulmonary disease (OR = 1.31 [1.23–1.40]), peptic ulcer disease (OR = 1.26 [1.17–1.35]), mild liver disease (OR = 1.27 [1.14–1.41]), depressive disorder (OR = 2.86 [2.70–3.02]), cerebrovascular disease (OR = 1.12 [1.10–1.20]), dementia (OR = 1.49 [1.26–1.78]), Parkinson’s disease’ (OR = 1.51 [1.22–1.88]), migraine (OR = 1.61 [1.44–1.82]), other-type headache (OR = 1.25 [1.03–1.52]), shoulder arthritis (OR = 1.15 [1.06–1.26]), knee arthritis (OR = 1.11 [1.04–1.18]), and ankle arthritis (OR = 1.32 [1.17–1.48]). All the results from the main statistical analysis are presented in Appendix A.

### 3.6. Validation of Risk Factors: Sensitivity Analysis

During the study period, the annual prevalence of sleep disorder in the year before the index surgery (main analysis) was similar to the proportions of patients who used sleep medication for over four weeks during the 90 days before the index surgery (Table 1): 5.3% vs. 5.2% in 2016, 5.4% vs. 5.4% in 2017, and 5.8% vs. 5.8% in 2018. Therefore, the target outcome for the sensitivity analysis was determined as the use of sleep medication for over four weeks during the 90 days before the index surgery. Except for region of residence and other-type headaches, most variables in the main analysis remained significant in the sensitivity analysis (Table 7). In addition, congestive heart failure, uncomplicated diabetes, and renal disease, including end-stage renal disease, were newly identified as significant variables in the sensitivity analysis. All the results from the sensitivity analysis are presented in Appendix A.

### 3.7. Validation of Estimates: Bootstrap Sampling

In the main analysis, the relative bias of the estimates for the risk factors was very low at between −4.45 and 2.21%, except for that of cerebrovascular disease (−16%). In the sensitivity analysis, the relative bias of the estimates was also very low between −5.13 and 6.99%. Bootstrap-adjusted odds ratios and 95% confidence intervals for the multivariable model are also displayed in Figure 4 (main analysis) and Figure 5 (sensitivity analysis). Multicollinearity among covariates was low, and all variance inflation factors were less than 1.9.

## 4. Discussions

To the best of our knowledge, this is the largest study to investigate the epidemiology of preoperative sleep disturbance in patients who underwent surgery for degenerative spinal disease. Among the 106,837 patients, the prevalence of sleep disorder was 5.5% (*n* = 5847), and during the 90 days before the spinal surgery, sleep medication was used over four weeks in 5.5% of the cohort (*n* = 5864) and over eight weeks in 3.8% (*n* = 4009) of the cohort. The prevalence of sleep disturbance differed according to the spinal regions, and sleep disorder was present in 6.9%, 5.7%, and 4.4% of patients with thoracic, lumbar, and cervical lesions, respectively. However, the spinal region was not a significant risk factor for sleep disorders in the multivariable analysis (Appendix A). The presence of sleep disorder in patients who underwent surgery for degenerative spinal disease was significantly associated with the following factors: Older age; female sex; urban residence; surgery at a tertiary hospital; general comorbidities, including peripheral vascular disease, chronic pulmonary disease, peptic ulcer disease, and mild liver disease; neuropsychiatric disorders, including depressive disorder, cerebrovascular disease, dementia, Parkinson’s disease, migraine, and other-type headache; and arthritis of the shoulder, knee, and ankle joints.

Compared with the prevalence of sleep disturbance in recent studies in the general population (1.6 to 18.6%) [23], and in patients with degenerative spinal disease (11 to 74%) [12,13,14,15,16,17], the prevalence of sleep disturbance in our cohort (3.8 to 5.5%, Table 3) is quite low. This difference results from the different methods used to evaluate sleep disturbance. Most previous studies used self-administered questionnaire-based surveys without objective clinical evidence to evaluate sleep disturbance, and the prevalence could have been overestimated. In contrast, in our study, sleep disturbance was only defined as present when the sleep disorder was diagnosed by doctors after a hospital visit or when sleep medication was prescribed for a sufficient period. Therefore, the prevalence of sleep disturbance in our cohort could have been underestimated.

The core results of our analysis identifying the independent factors associated with sleep disturbance are presented in Figure 4. In Figure 4, the bootstrap-adjusted ORs and 95% confidence intervals of individual factors can be evidently divided into four groups: (1) Age, (2) other demographic factors and general comorbidities, (3) neuropsychiatric disorders, and (4) osteoarthritis of the extremities. While older age is a strong risk factor for sleep disturbance in our cohort, other demographic variables including sex and region of residence, various general comorbidities, and osteoarthritis of the extremities did not show comparable risks for sleep disturbance (all their adjusted ORs are below 1.4). In contrast, most neuropsychiatric disorders showed higher ORs for sleep disturbance than general comorbidities, and depressive disorder was the most prominent risk factor for sleep disturbance (OR = 2.86 [2.72–3.00]).

Interestingly, the prevalence of sleep disturbance differed according to the location of the spinal lesion (Figure 3), and univariable analysis identified significant differences according to spinal regions, especially between the cervical and lumbar regions (*p* < 0.001, Appendix A). However, the location of the spinal lesion was not an independent risk factor for sleep disturbance in the multivariable analysis (Table 6 and Table 7). Based on the results of our study, we suggest that regional differences in the prevalence of sleep disturbance in the unadjusted analysis (Figure 3 and Appendix A) result from regional differences in factors associated with sleep disturbance, such as neuropsychiatric disorders (Table 4) and degenerative joint diseases of the extremities (Table 5).

The major advantage of our study is that we could precisely present the prevalence of sleep disturbance according to four groups of factors (Table 2, Table 3, Table 4 and Table 5). Our database represents the entire Korean population, and these prevalence rates can be used as the base rates for sleep disturbance in patients with specific risk factors. It is well known that the accuracy of prediction by a simple ‘base rate’ of the entire population can be comparable to that obtained from a complex statistical analysis [24]. Although our prediction model (Table 6 and Table 7) for sleep disturbance could be inevitably biased by unknown confounders due to the study’s limitations, our prevalence rates of sleep disturbance presented by four groups of factors can be used as a reasonable source of the base rates.

This study has some limitations. First, the HIRA database is a claims database not originally designed for clinical research. Although we used validated data retrieval methods for the HIRA database, possible discrepancies between the diagnostic codes in the database and the actual diseases may be potential sources of bias. However, the HIRA system is based on our compulsory national health insurance system, and the control policy for high-revenue spinal surgeries has been the object of priority. Therefore, therapeutic information about drug and device use, as well as precise surgical approaches, is thoroughly reviewed by government officials and is thus considered very accurate. Second, information possibly related to sleep disturbance, including the radiologic degree of spinal degeneration such as disc degeneration or canal stenosis, or the degree of neurological impairment, could not be included in the study. In particular, information regarding the radiologic degree or types of degeneration could have influenced our results as a confounder [12,13], although most patients who underwent surgical treatment have an end-stage degenerative spinal disease. To reduce the influence of such unknown confounders, we performed a two-step validation procedure, and the results were consistent. Third, we could not include patients with degenerative spinal deformities because of the limited data capacity for analysis. Finally, we particularly focused on investigating the sleep disturbance according to spinal regions, and multivariable analysis showed that the prevalence of sleep disturbance was not significantly different among spinal regions. However, due to the lack of important information, including the presence of various symptoms or signs depending on spinal regions and their severity, our results could be biased. Previous studies have suggested different mechanisms of sleep disturbance according to spinal regions, and further studies including such important clinical information would be interesting and helpful to understand the actual mechanisms of sleep disturbance in patients with degenerative spinal disease.

In conclusion, our population-based study using a nationwide database identified that the prevalence of sleep disturbance in patients undergoing surgery for degenerative spinal disease was 5.5% (5847 of 106,837 patients). Although the prevalence of sleep disturbance differed according to spinal regions, the spinal region was not a significant risk factor for sleep disorder in the multivariable analysis. In addition, we identified four groups of independent risk factors: (1) Age, (2) other demographic factors and general comorbidities, (3) neuropsychiatric disorders, and (4) osteoarthritis of the extremities. Our results, including the prevalence rates of sleep disturbance based on the entire population and the identified risk factors, provide clinicians with a reasonable reference for evaluating sleep disturbance in patients with degenerative spinal diseases and future research.

## Figures and Tables

**Figure 1 jcm-11-05932-f001:**
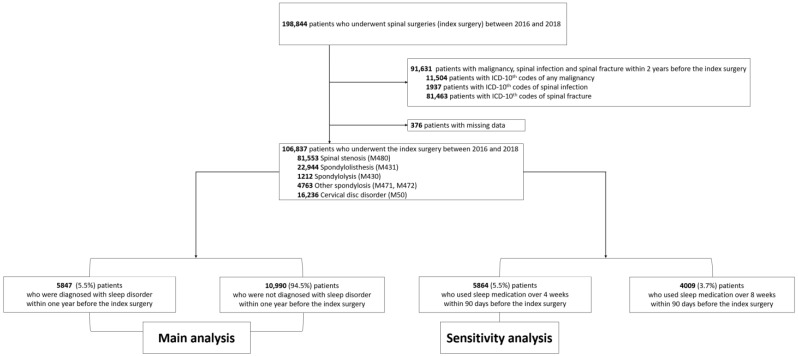
Enrollment of study patients.

**Figure 2 jcm-11-05932-f002:**
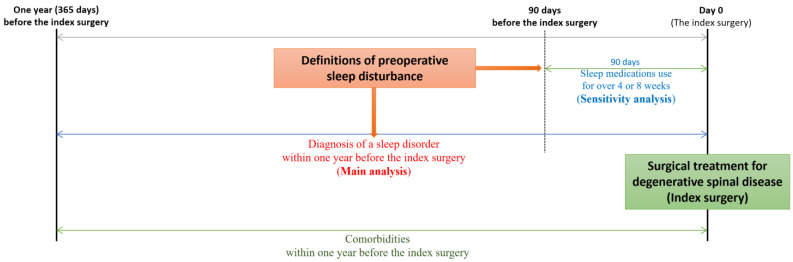
Definitions of sleep disturbance in the main and sensitivity analyses. The term “sleep disorder” has been used when sleep problems were identified using International Classification of Diseases, tenth revision (ICD-10) codes alone. The term “sleep disturbance” has been used when sleep problems were identified using the following two criteria: Diagnosis of a sleep disorder using ICD-10 codes and the use of sleep medication.

**Figure 3 jcm-11-05932-f003:**
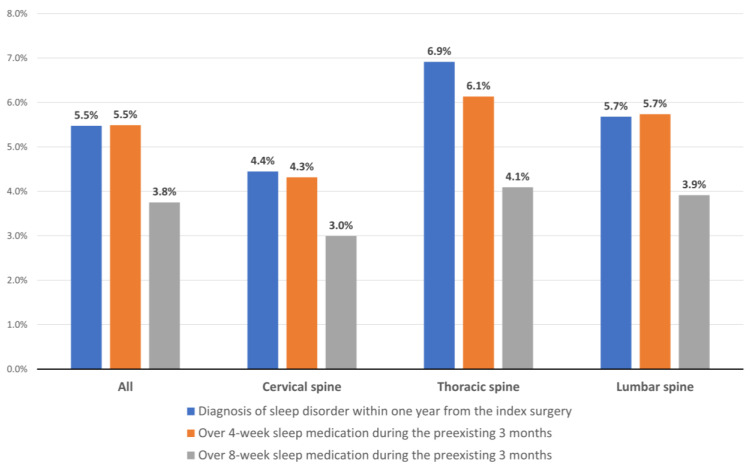
Prevalence of sleep disturbance by spinal region according to the three definitions.

**Figure 4 jcm-11-05932-f004:**
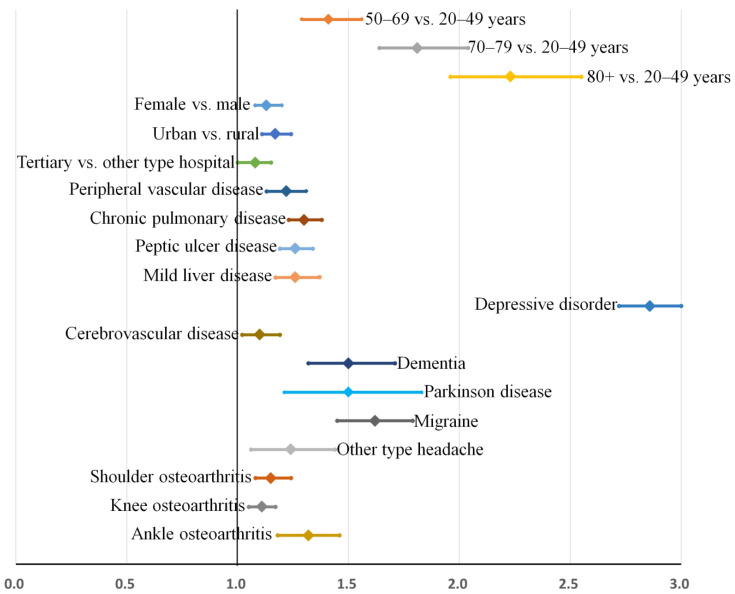
Risk factors for sleep disorder (main analysis). Bootstrap-adjusted odds ratios and their 95% confidence intervals have been presented. Risk factors can be categorized into four groups: (1) Age, (2) other demographic factors and general comorbidities, (3) neuropsychiatric disorders, and (4) osteoarthritis of the extremities.

**Figure 5 jcm-11-05932-f005:**
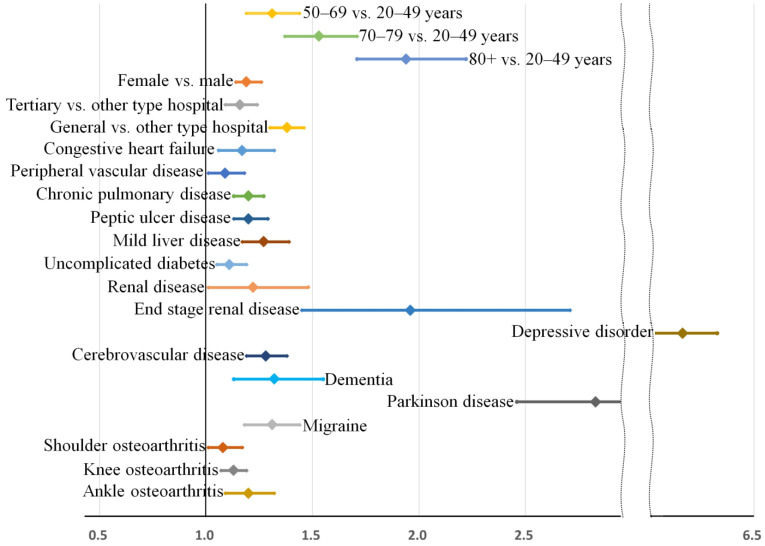
Risk factors for sleep medication use for over 8 weeks during the preoperative 90 days (subgroup analysis). Bootstrap-adjusted odds ratios and their 95% confidence intervals have been presented. Risk factors can be categorized into four groups: (1) Age, (2) other demographic factors and general comorbidities, (3) neuropsychiatric disorders, and (4) osteoarthritis of the extremities.

**Table 1 jcm-11-05932-t001:** Annual prevalence of sleep disturbance according to the three definitions.

Year	Spinal Surgery (*n*)	Patients Diagnosed with Sleep Disorder within One Year from the Index Surgery			Prevalence According to Sleep Medication during the Preexisting 90 Days					
					Over 4-Week Sleep Medication			Over 8-Week Sleep Medication		
		(*n*)	Incidence (%)	95% CI	(*n*)	Incidence (%)	95% CI	(*n*)	Incidence (%)	95% CI
2016	35,507	1866	5.3%	[5.0–5.5]	1839	5.2%	[4.9–5.4]	1229	3.5%	[3.3–3.7]
2017	35,459	1912	5.4%	[5.2–5.6]	1932	5.4%	[5.2–5.7]	1319	3.7%	[3.5–3.9]
2018	35,871	2069	5.8%	[5.5–6.0]	2093	5.8%	[5.6–6.1]	1461	4.1%	[3.9–4.3]
All	106,837	5847	5.5%	[5.3–5.6]	5864	5.5%	[5.4–5.6]	4009	3.8%	[3.6–3.9]

**Table 2 jcm-11-05932-t002:** Prevalence of sleep disturbance according to the baseline characteristics.

Variables	Categories	All	Patients Diagnosed with Sleep Disorder within One Year from the Index Surgery		Prevalence according to Sleep Medication during the Preexisting 90 days			
					Over 4-Week Sleep Medication		Over 8-Week Sleep Medication	
Number of Patients		106,837	5847	5.5%	5864	5.5%	4009	3.8%
Age	Mean ± SD	62.9 ± 11.8	66.7 ± 10.5		66.9 ± 10.3		66.6 ± 10.4	
	20–49	14,014	378	2.7%	358	2.6%	266	1.9%
	50–69	58,533	2857	4.9%	2881	4.9%	2007	3.4%
	70–79	28,671	2115	7.4%	2116	7.4%	1393	4.9%
	80+	5619	497	8.8%	509	9.1%	343	6.1%
Sex	Male	51,242	2298	4.5%	2203	4.3%	1503	2.9%
	Female	55,595	3549	6.4%	3661	6.6%	2506	4.5%
Region	Urban	88,826	4953	5.6%	4903	5.5%	3323	3.7%
	Rural	18,011	894	5.0%	961	5.3%	686	3.8%
Hospital	Tertiary	18,442	1154	6.3%	1169	6.3%	814	4.4%
	General	20,772	1257	6.1%	1537	7.4%	1072	5.2%
	Others	67,623	3436	5.1%	3158	4.7%	2123	3.1%

**Table 3 jcm-11-05932-t003:** Prevalence of sleep disturbance according to comorbidities.

Variables	Categories	All	Patients Diagnosed with Sleep Disorder within One Year from the Index Surgery		Prevalence According to Sleep Medication during the Preexisting 90 Days			
					Over 4-Week Sleep Medication		Over 8-Week Sleep Medication	
Number of patients		106,837	5847	5.5%	5864	5.5%	4009	3.8%
Charlson comorbidity index score	Mean ± SD	1.14 ± 1.28	1.56 ± 1.44		1.67 ± 1.52		1.66 ± 1.54	
	0–2	75,632	4551	6.0%	4423	5.8%	3028	4.0%
	3–5	27,691	1195	4.3%	1310	4.7%	887	3.2%
	≥6	3514	101	2.9%	131	3.7%	94	2.7%
Comorbidities	Myocardial infarction	967	72	7.4%	73	7.5%	51	5.3%
	Congestive heart failure	3394	286	8.4%	314	9.3%	217	6.4%
	Peripheral vascular disease	12,062	969	8.0%	934	7.7%	644	5.3%
	Chronic pulmonary disease	24,116	1867	7.7%	1825	7.6%	1227	5.1%
	Rheumatologic disease	4010	292	7.3%	298	7.4%	198	4.9%
	Peptic ulcer disease	17,189	1341	7.8%	1331	7.7%	905	5.3%
	Liver disease							
	Mild	6686	485	7.3%	496	7.4%	341	5.1%
	Moderate to severe	83	7	8.4%	9	10.8%	5	6.0%
	Diabetes							
	Uncomplicated	23,105	1492	6.5%	1660	7.2%	1137	4.9%
	Complicated	6733	434	6.4%	559	8.3%	362	5.4%
	Hemiplegia or paraplegia	849	50	5.9%	70	8.2%	42	4.9%
	Renal disease	2053	179	8.7%	211	10.3%	157	7.6%
	End stage renal disease	379	39	10.3%	57	15.0%	39	10.3%
	Osteoporosis	15,495	1185	7.6%	1189	7.7%	813	5.2%
Concurrent neuropsychiatric disorders	Depressive disorder	23,921	2818	11.8%	3740	15.6%	2806	11.7%
	Cerebrovascular disease	9502	808	8.5%	971	10.2%	695	7.3%
	Dementia	1388	167	12.0%	160	11.5%	109	7.9%
	Parkinson disease	875	100	11.4%	175	20.0%	152	17.4%
	Migraine	3222	384	11.9%	356	11.0%	242	7.5%
	Tension type headache	3011	343	11.4%	329	10.9%	219	7.3%
	Other-type headache	4304	469	10.9%	449	10.4%	303	7.0%
Concurrent osteoarthritis of extremities	Shoulder	8503	674	7.9%	648	7.6%	450	5.3%
	Elbow	2276	141	6.2%	121	5.3%	86	3.8%
	Wrist	2268	183	8.1%	192	8.5%	135	6.0%
	Hip	7104	542	7.6%	531	7.5%	357	5.0%
	Knee	24,338	1828	7.5%	1898	7.8%	1274	5.2%
	Ankle	4024	368	9.1%	353	8.8%	239	5.9%

**Table 4 jcm-11-05932-t004:** Prevalence of sleep disturbance according to spinal regions and concurrent neuropsychiatric disorders.

Spinal Regions	According to Concurrent Neuropsychiatric Disorders	Cases (*n*) with Its Proportion (%)		Patients Diagnosed with Sleep Disorder within one Year from the Index Surgery		Prevalence According to Sleep Medication during the Preexisting 90 Days			
						Over 4-Week Sleep Medication		Over 8-Week Sleep Medication	
Cervical	All cases	18,819	(100)	837	4.4%	812	4.3%	563	3.0%
	Depressive disorder	3660	19.4%	372	10.2%	526	14.4%	403	11.0%
	Cerebrovascular disease	1380	7.3%	107	7.8%	136	9.9%	97	7.0%
	Dementia	103	0.5%	10	9.7%	14	13.6%	11	10.7%
	Parkinson disease	88	0.5%	8	9.1%	20	22.7%	17	19.3%
	Migraine	566	3.0%	60	10.6%	54	9.5%	37	6.5%
	Tension type headache	513	2.7%	49	9.6%	45	8.8%	29	5.7%
	Other-type headache	742	3.9%	71	9.6%	59	8.0%	37	5.0%
Thoracic	All cases	1027	(100)	71	6.9%	63	6.1%	42	4.1%
	Depressive disorder	271	26.4%	30	11.1%	36	13.3%	24	8.9%
	Cerebrovascular disease	127	12.4%	16	12.6%	12	9.4%	7	5.5%
	Dementia	18	1.8%	1	5.6%	0	0.0%	0	0.0%
	Parkinson disease	6	0.6%	1	16.7%	2	33.3%	2	33.3%
	Migraine	24	2.3%	2	8.3%	3	12.5%	2	8.3%
	Tension type headache	31	3.0%	3	9.7%	2	6.5%	1	3.2%
	Other-type headache	45	4.4%	3	6.7%	3	6.7%	1	2.2%
Lumbar	All cases	86,991	(100)	4939	5.7%	4989	5.7%	3404	3.9%
	Depressive disorder	19,990	23.0%	2416	12.1%	3178	15.9%	2379	11.9%
	Cerebrovascular disease	7995	9.2%	685	8.6%	823	10.3%	591	7.4%
	Dementia	1267	1.5%	156	12.3%	146	11.5%	98	7.7%
	Parkinson disease	781	0.9%	91	11.7%	153	19.6%	133	17.0%
	Migraine	2632	3.0%	322	12.2%	299	11.4%	203	7.7%
	Tension type headache	2467	2.8%	291	11.8%	282	11.4%	189	7.7%
	Other-type headache	3517	4.0%	395	11.2%	387	11.0%	265	7.5%

**Table 5 jcm-11-05932-t005:** Prevalence of sleep disturbance according to concurrent osteoarthritis of extremities.

Spinal Regions	Categories	Extremities	Cases (*n*) with ItsProportion		Patients Diagnosed with Sleep Disorder within One Year from the Index Surgery		Prevalence According to Sleep Medication during the Preexisting 90 Days			
							Over 4-Week Sleep Medication		Over 8-Week Sleep Medication	
Cervical	All cases		18,819	(100)	837	4.4%	812	4.3%	563	3.0%
	Upper extremities	Shoulder	2214	11.8%	156	7.0%	142	6.4%	101	4.6%
		Elbow	568	3.0%	25	4.4%	24	4.2%	18	3.2%
		Wrist	532	2.8%	39	7.3%	42	7.9%	27	5.1%
	Lower extremities	Hip	446	2.4%	38	8.5%	45	10.1%	29	6.5%
		Knee	2283	12.1%	155	6.8%	173	7.6%	123	5.4%
		Ankle	459	2.4%	46	10.0%	38	8.3%	31	6.8%
Thoracic	All cases		1027	(100)	71	6.9%	63	6.1%	42	4.1%
	Upper extremities	Shoulder	94	9.2%	8	8.5%	4	4.3%	2	2.1%
		Elbow	27	2.6%	3	11.1%	3	11.1%	2	7.4%
		Wrist	25	2.4%	2	8.0%	3	12.0%	2	8.0%
	Lower extremities	Hip	94	9.2%	8	8.5%	10	10.6%	5	5.3%
		Knee	334	32.5%	31	9.3%	34	10.2%	23	6.9%
		Ankle	68	6.6%	6	8.8%	9	13.2%	8	11.8%
Lumbar	All cases		86,991	(100)	4939	5.7%	4989	5.7%	3404	3.9%
	Upper extremities	Shoulder	6195	7.1%	510	8.2%	502	8.1%	347	5.6%
		Elbow	1681	1.9%	113	6.7%	94	5.6%	66	3.9%
		Wrist	1711	2.0%	142	8.3%	147	8.6%	105	6.1%
	Lower extremities	Hip	6564	7.5%	496	7.6%	476	7.3%	323	4.9%
		Knee	21,721	25.0%	1642	7.6%	1691	7.8%	1128	5.2%
		Ankle	3497	4.0%	316	9.0%	306	8.8%	200	5.7%

**Table 6 jcm-11-05932-t006:** Risk factors for sleep disorder: Main analysis.

Variables	Categories	Model 1 (Univariable)		Model 2 (Fully Adjusted)		Model 3 (Bootstrap Validation after Fully Adjusted)	
		Odds Ratio (95% Confidence Interval)	*p*-Value	Adjusted Odds Ratio (95% Confidence Interval)	*p*-Value	Adjusted Odds Ratio (95% Confidence Interval)	Relative Bias (%)
Age	50–69 vs. 20–49 years	1.85 [1.66–2.06]	<0.001	1.40 [1.25–1.57]	<0.001	1.41 [1.29–1.56]	2.21
	70–79 vs. 20–49 years	2.87 [2.57–3.21]	<0.001	1.80 [1.60–2.03]	<0.001	1.81 [1.64–2.04]	0.89
	80+ vs. 20–49 years	3.50 [3.05–4.02]	<0.001	2.22 [1.92–2.58]	<0.001	2.23 [1.96–2.55]	0.80
Sex	Female vs. male	1.45 [1.38–1.53]	<0.001	1.14 [1.07–1.21]	<0.001	1.13 [1.08–1.20]	−4.45
Region	Urban vs. rural	1.13 [1.05–1.22]	0.001	1.18 [1.09–1.27]	<0.001	1.17 [1.11–1.24]	−2.97
Hospital	Tertiary vs. others	1.25 [1.16–1.34]	<0.001	1.08 [1.00–1.16]	0.047	1.08 [1.00–1.15]	−3.33
Comorbidities	Peripheral vascular disease	1.61 [1.50–1.73]	<0.001	1.22 [1.13–1.32]	<0.001	1.22 [1.13–1.31]	0.68
	Chronic pulmonary disease	1.66 [1.57–1.76]	<0.001	1.31 [1.23–1.40]	<0.001	1.30 [1.23–1.38]	−1.42
	Peptic ulcer disease	1.60 [1.50–1.70]	<0.001	1.26 [1.17–1.35]	<0.001	1.26 [1.19–1.34]	−0.91
	Mild liver disease	1.38 [1.26–1.52]	<0.001	1.27 [1.14–1.41]	<0.001	1.26 [1.17–1.37]	−1.68
Comorbidities associated neuropsychiatric disorders	Depressive disorder	3.52 [3.34–3.72]	<0.001	2.86 [2.70–3.02]	<0.001	2.86 [2.72–3.00]	0.03
	Cerebrovascular disease	1.70 [1.58–1.84]	<0.001	1.12 [1.10–1.20]	0.040	1.10 [1.02–1.19]	−16.00
	Dementia	2.41 [2.04–2.83]	<0.001	1.49 [1.26–1.78]	<0.001	1.50 [1.32–1.71]	0.96
	Parkinson disease	2.25 [1.83–2.78]	<0.001	1.51 [1.22–1.88]	<0.001	1.50 [1.21–1.83]	−1.63
	Migraine	2.43 [2.18–2.71]	<0.001	1.61 [1.44–1.82]	<0.001	1.62 [1.45–1.79]	1.76
	Other-type headache	2.21 [2.00–2.44]	<0.001	1.25 [1.03–1.52]	0.023	1.24 [1.06–1.44]	−2.75
Concurrent osteoarthritis	Shoulder	1.55 [1.43–1.69]	<0.001	1.15 [1.06–1.26]	0.002	1.15 [1.08–1.24]	1.17
	Knee	1.59 [1.50–1.68]	<0.001	1.11 [1.04–1.18]	0.002	1.11 [1.05–1.17]	−1.40
	Ankle	1.79 [1.60–2.00]	<0.001	1.32 [1.17–1.48]	<0.001	1.32 [1.18–1.46]	0.67

Relative bias was estimated as the difference between the mean bootstrapped regression coefficient estimates (model 3) and the mean parameter estimates of multivariable model (model 2) divided by the mean parameter estimates of multivariable model (model 2).

**Table 7 jcm-11-05932-t007:** Risk factors for over 4-week sleep medication during the preoperative 90 days: Sensitivity analysis.

Variables	Categories	Univariable		Model 2 (Fully Adjusted)		Model 3 (Bootstrap Validation after Fully Adjusted)	
		Odds Ratio (95% Confidence Interval)	*p*-Value	Adjusted Odds Ratio (95% Confidence Interval)	*p*-Value	Adjusted Odds Ratio (95% Confidence Interval)	Relative Bias (%)
Age	50–69 vs. 20–49 years	1.97 [1.77–2.21]	<0.001	1.32 [1.17–1.49]	<0.001	1.31 [1.19–1.44]	−2.60
	70–79 vs. 20–49 years	3.04 [2.71–3.41]	<0.001	1.54 [1.36–1.75]	<0.001	1.53 [1.37–1.71]	−0.78
	80+ vs. 20–49 years	3.80 [3.31–4.37]	<0.001	1.95 [1.68–2.27]	<0.001	1.94 [1.71–2.22]	−0.69
Sex	Female vs. male	1.57 [1.49–1.66]	<0.001	1.20 [1.13–1.27]	<0.001	1.19 [1.14–1.26]	−3.96
Hospital	Tertiary vs. others	1.38 [1.29–1.48]	<0.001	1.17 [1.08–1.15]	<0.001	1.16 [1.09–1.24]	−4.46
	General vs. others	1.63 [1.53–1.74]	<0.001	1.38 [1.29–1.47]	<0.001	1.38 [1.30–1.46]	−0.34
Comorbidities	Congestive heart failure	1.80 [1.60–2.03]	<0.001	1.16 [1.02–1.33]	0.023	1.17 [1.06–1.32]	6.99
	Peripheral vascular disease	1.53 [1.42–1.65]	<0.001	1.09 [1.00–1.18]	0.040	1.09 [1.01–1.18]	−0.67
	Chronic pulmonary disease	1.60 [1.51–1.69]	<0.001	1.21 [1.13–1.29]	<0.001	1.20 [1.13–1.27]	−4.26
	Peptic ulcer disease	1.58 [1.48–1.68]	<0.001	1.20 [1.11–1.29]	<0.001	1.20 [1.13–1.29]	−0.34
	Mild liver disease	1.42 [1.29–1.56]	<0.001	1.27 [1.14–1.40]	<0.001	1.27 [1.17–1.39]	−0.13
	Uncomplicated diabetes	1.46 [1.38–1.55]	<0.001	1.12 [1.04–1.20]	0.002	1.11 [1.05–1.19]	−4.17
	Renal disease	2.01 [1.74–2.32]	<0.001	1.23 [1.01–1.49]	0.042	1.22 [1.01–1.48]	−5.03
	End stage renal disease	3.07 [2.31–4.07]	<0.001	1.97 [1.39–2.79]	<0.001	1.96 [1.45–2.71]	−0.76
Comorbidities associated neuropsychiatric disorders	Depressive disorder	7.05 [6.67–7.45]	<0.001	5.84 [5.51–6.18]	<0.001	5.84 [5.57–6.16]	0.01
	Cerebrovascular disease	2.15 [2.00–2.31]	<0.001	1.28 [1.18–1.39]	<0.001	1.28 [1.19–1.38]	0.54
	Dementia	2.28 [1.93–2.69]	<0.001	1.33 [1.11–1.59]	0.002	1.32 [1.13–1.55]	−1.68
	Parkinson disease	4.41 [3.73–5.21]	<0.001	2.80 [2.34–3.36]	<0.001	2.83 [2.46–3.32]	1.00
	Migraine	2.21 [1.98–2.48]	<0.001	1.30 [1.15–1.47]	<0.001	1.31 [1.18–1.44]	1.88
Concurrent osteoarthritis	Shoulder	1.47 [1.35–1.60]	<0.001	1.08 [1.02–1.17]	0.013	1.08 [1.01–1.17]	3.82
	Knee	1.68 [1.58–1.77]	<0.001	1.14 [1.06–1.21]	<0.001	1.13 [1.07–1.19]	−5.13
	Ankle	1.70 [1.52–1.90]	<0.001	1.19 [1.06–1.35]	0.004	1.20 [1.09–1.32]	5.14

Relative bias was estimated as the difference between the mean bootstrapped regression coefficient estimates (model 3) and the mean parameter estimates of multivariable model (model 2) divided by the mean parameter estimates of multivariable model (model 2).

## Data Availability

The datasets generated for the current study are not publicly available due to Data Protection Laws and Regulations in Korea, but the analyzing results are available from the corresponding authors on reasonable request.

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
