# Peer review of "Prevalence of Sleep Disturbance and Its Risk Factors in Patients Who Undergo Surgical Treatment for Degenerative Spinal Disease: A Nationwide Study of 106,837 Patients"

_jcm, 2022, doi:10.3390/jcm11195932_

Round 1
Reviewer 1 Report
Authors investigated the prevalence of sleep disturbance and its risk factors in patients who undergo surgical treatment for degenerative spinal disease, and found that the prevalence of sleep disturbance was 5.5% and found several risk factors for sleep disturbance. Although authors discussed an interesting topic, I have some concerns in this study. My comments are as follows.
1. Why did authors define the sleep disturbance as shown in the manuscript? Was it comparable to other studies? This is important for discussing epidemiology. Also, please show the references of the prevalence of sleep disturbance in a general population.
2. Please add more discussion on why the prevalence of sleep disturbance was different in different spinal lesions.
3. Sometimes the words of sleep disturbance and sleep disorder seem to be confusing and misused.
4. In abstract, the definition of “sleep disorder” should be described.
5. The word “cluster” would not be appropriate because this study did not perform the cluster analyses.
6. There were some redundant parts including Results section, so please refine the text.
Reviewer 2 Report
Epidemiological studies on sleep disturbances in patients with spinal disease are of great interest to spine surgeons. I read the article with great interest.
I do not disagree with your findings and conclusions, but I believe there are many issues with the way the paper is structured and described. I hope that the attached file will help you to correct them.
While I understand the desire to present all of the evaluated content, I feel that the repetition of the same content in the text, tables, and figures hinders the reader's understanding of this paper.

Reviewer 3 Report
This study is significant for its size and the rigor of its statistical analysis. There is, however, one major flaw. The definition of "sleep disorders". This vague definition does not allow us to draw any conclusions on the results. Did they have more obstructive sleep apnea, did they have more chronic insomnia, Restless Legs Syndrome? Which disorder. This has not been outlined. Also the prevalence of all sleep disorders among their cohort seems very low. Could it be that the charts were not thoroughly reviewed? Who had made the diagnosis? Is there a likelihood that some never went to a sleep doctor?
Round 2
Reviewer 1 Report
The manuscript has been revised well according to the reviewer’s comments.
Author Response
Response to the first reviewer’s comments
The manuscript has been revised well according to the reviewer’s comments.
-> Thank you very much for your previous comments regarding our study.
Reviewer 2 Report
I found it easier to understand than the first edition. Thank you for the commentary on the content. However, I still have trouble understanding some parts, so let me ask you a question.
About Disscusions
(1) [the spinal region was not a significant risk factor for sleep disorders in the multivariable analysis. ]
I think this content is important for your paper, but there is no indication of it in the tables and graphs. Where else is it indicated other than in the appendix? Or do you mean that there is no significant difference after a statistical comparison of the prevalence by site in Figure 3?
(2) [Interestingly, the prevalence of sleep disturbance differed according to the location of the spinal lesionlesions (Figure 34). However, the location of the spinal lesionit was not an independent risk factor for sleep disturbance in the multivariable analysis (Table 6 and 7). ]
Looking at Tables 6 and 7, there is no mention of a spinal site anywhere. Does the lack of mention mean that it is not a risk factor?
(3)The purpose of this paper is as stated in the introduction, but the hidden purpose of this paper is to examine whether sleep disturbances differ depending on the site of the spinal cord lesion. This point seems to make it difficult for the reader to understand.(I feel that it would be easier to understand the content if the paper were originally divided into two parts.) Alternatively, it may be better to state the above as a third objective.
From a clinician's point of view, it is difficult to understand why the results of this study show differences in symptoms depending on the site of the spinal cord lesion. I am interested in how symptoms caused by myelopathy and cauda equina disorder affect sleep. However, since there is no data on symptoms in the present study, I do not think that this has been verified. I think it would be a good idea to mention this point. (I do not disagree with the risk factors found in the results of this study, as the effects of sleep disorders are multifactorial and I think the methodology used is appropriate.)
Reviewer 3 Report
It has improved sufficiently.
English editing required.
Author Response
Response to the third reviewer’s comments
It has improved sufficiently.
English editing required.
-> Thank you very much for your previous comments regarding our manuscript. According to your comment, we have revised the English of our manuscript using a professional language editing service.